# Inner core composition paradox revealed by sound velocities of Fe and Fe-Si alloy

Haijun Huang [ID] [1], Lili Fan[1], Xun Liu[1], Feng Xu [ID] [1], Ye Wu [ID] [1], Gang Yang [ID] [1], Chunwei Leng[1], Qingsong Wang[2], Jidong Weng[2], Xiang Wang[2], Lingcang Cai[2] & Yingwei Fei [ID] [3✉]

Knowledge of the sound velocity of core materials is essential to explain the observed anomalously low shear wave velocity ($V_S$) and high Poisson's ratio ($\sigma$) in the solid inner core. To date, neither $V_S$ nor $\sigma$ of Fe and Fe-Si alloy have been measured under core conditions. Here, we present $V_S$ and $\sigma$ derived from direct measurements of the compressional wave velocity, bulk sound velocity, and density of Fe and Fe-8.6 wt%Si up to ~230 GPa and ~5400 K. The new data show that neither the effect of temperature nor incorporation of Si would be sufficient to explain the observed low $V_S$ and high $\sigma$ of the inner core. A possible solution would add carbon (C) into the solid inner core that could further decrease $V_S$ and increase $\sigma$. However, the physical property-based Fe-Si-C core models seemingly conflict with the partitioning behavior of Si and C between liquid and solid Fe.

---

[1] School of Science, Wuhan University of Technology, Wuhan, Hubei 430070, China. [2] National Key Laboratory of Shock Wave and Detonation Physics, Institute of Fluid Physics, China Academy of Engineering Physics, Mianyang, Sichuan 621900, China. [3] Earth and Planets Laboratory, Carnegie Institution for Science, Washington, DC 20015, USA. ✉email: yfei@carnegiescience.edu

Earth's core, containing a liquid outer core and a solid inner core, constitutes 32% of the mass of the planet, and its composition strongly influences the heat flow in Earth's deep interior, the crystallization of the inner core, and the evolution of the magnetic field[1]. High-pressure experiments and geophysical observations have revealed that the iron-nickel outer core and inner core might contain ~10 wt% and ~4 wt% light elements, respectively[2–4]. The proposed light elements include carbon (C), hydrogen (H), oxygen (O), sulfur (S), and silicon (Si), but the identity and relative amounts of the light elements are still debated. Evidence from geochemistry[5], metal–silicate partitioning experiments[6,7], and isotope fractionation experiments[8,9] supports Si as the dominant light element. C might also be present in the Earth's core due to its abundance in primitive chondritic meteorites[10], and its siderophile nature during metal-silicate differentiation[11–13]. Any proposed composition models of the core must simultaneously satisfy the core density ($\rho$), the bulk sound velocity ($V_B$) of the liquid outer core, and the compressional wave velocity ($V_P$) and shear wave velocity ($V_S$) of the solid inner core, defined by seismic observations[14,15]. However, it has been challenging to measure $V_P$ at pressure ($P$) and temperature ($T$) relevant to the Earth's core conditions, and even more challenging for the measurements of $V_S$. Hence, measurements on core materials need to be extrapolated to core conditions to compare them with the seismic observations.

The extrapolation of $V_P$ is usually made according to Birch's law[2], which assumes a linear relationship between $\rho$ and $V_P$. It is still debated whether Birch's law holds under high $P$-$T$ conditions. For example, static data by inelastic X-ray scattering (IXS) showed that the $V_P$-$\rho$ relation for iron[16] follows Birch's law up to 93 GPa and 1100 K. On the other hand, measurements by IXS[17,18] and nuclear-resonant inelastic X-ray scattering (NRIXS)[19] showed that $V_P$ decreases with temperature (<3000 K). The static data combined with the early shockwave data[20] indicate that Birch's law does not hold at moderate and high temperature. The disagreement on Birch's law would significantly affect the estimation of the amounts of Si in the inner core depending on models of the temperature effect on $V_P$, ranging from 1–2 wt%[21], and 3–6 wt%[22], to 8 wt%[17].

For $V_S$ of iron alloys, most of the data were derived from NRIXS measurements at high pressure and room temperature[23–26]. Estimate of the temperature effect at moderate and high temperature may be doubtful because the extraction of the phonon density of state is from the NRIXS spectra based on a quasi-harmonic model[16]. Therefore, no reliable $V_S$ data of iron alloys are available under the Earth's core conditions. Comparison of the experimental $V_S$ measurements with the seismic observations indicates that the extrapolated $V_S$ of iron at 300 K is approximately two times of the inner core value, and the extrapolated Poisson's ratio ($\sigma$) of iron is approximately half of that observed in the inner core[23]. High temperature and the presence of certain light elements such as C[27,28] in the Earth's inner core are expected to decrease the $V_S$ value and increase the $\sigma$ value to match the observations. To model the $V_S$ profile of the inner core, we need to determine the temperature and compositional dependence of $V_S$ of iron alloys under high $P$-$T$ conditions relevant to the Earth's inner core.

Computer simulations are capable of calculating $\rho$, $V_P$, and $V_S$ under $P$-$T$ conditions in the core, but the simulated results have not reached a consensus on the temperature dependence of Birch's law for $V_P$-$\rho$ and $V_S$ of iron. For example, the simulated results considering the effect of anharmonicity[29] revealed that both $V_P$ and $V_S$ of iron decreased with temperature at constant density and could match the seismic data of the inner core. On the other hand, more recent calculations[30,31] showed that both $V_P$ and $V_S$ of iron increased almost linearly with density between

0~5500 K, and proposed that the $\rho$, $V_P$ and $V_S$ of $Fe_{60}Si_2C_2$ (with 1.6 wt% Si and 0.7 wt% C) at 360 GPa and 6500 K could match the seismic data when considering the pre-melting effect[31]. However, all simulated $V_P$ and $V_S$ of iron at 0 K were higher than the extrapolation of experimental data at 300 K.

From the experimental measurements and theoretical calculations of $V_P$ and $V_S$ reported so far, there is no consensus on the temperature dependence of $V_P$ and $V_S$ of iron and iron alloys at high pressure and temperature (Fig. 1). Here, we report new direct measurements of $V_P$ and $V_B$ of Fe and Fe–Si alloy (8.6 wt% Si) by shock compression to investigate the temperature dependence on $V_P$, $V_S$ and $\sigma$ under conditions relevant to the Earth's core. The results provide a test of Birch's law at simultaneous high pressure and temperature and determine if Si is a viable light element in the inner core.

## Results

**Sound velocity determination.** Using the reverse-impact technique (RIT)[32,33] and optical analyzer technique (OAT)[20,33], the sound velocities of Fe and Fe-8.6 wt% Si (hereafter Fe-8.6Si) were measured with a two-stage light gas gun. The experimental setup and the Lagrangian wave propagation diagram for the RIT are depicted in Supplementary Fig. 1. The data processing method is described in the Methods section. The RIT signals for Fe and Fe-8.6Si are shown in Supplementary Fig. 2. The RIT experiments produce both $V_P$ and $V_B$ measurements. To improve the accuracy, we take the derivative of the particle velocity with respect to time to pinpoint the arrival time of the rarefaction wave and the elastic-plastic transition point (Supplementary Fig. 3). Supplementary Table 1 lists the direct measurements of $V_P$ and $V_B$. Then we obtained the shear wave velocity via $V_S = [3(V_P^2 - V_B^2)/4]^{1/2}$ and the Poisson's ratio via $\sigma = 0.5(V_P^2 - 2V_S^2)/(V_P^2 - V_S^2)$.

Because of the low impedance of the lithium fluoride (LiF) window, the maximal pressure reached by the RIT is limited to ~160 GPa. For higher pressures, the $V_P$ was obtained by the OAT, which utilizes multiple samples with different thicknesses to determine the catch-up thickness for accurate measurements of the sound velocity. The experimental setup and Lagrangian wave propagation diagram for the OAT are illustrated in Supplementary Fig. 4. Supplementary Fig. 5 shows representative signals and the determined catch-up thickness of experiment 180102 at 126 GPa. Similarly, we also take the derivative of the particle velocity with respect to time to accurately determine the arrival time of the rarefaction wave at the sample/window interface. Supplementary Table 2 lists the measured $V_P$ along with the Hugoniot parameters. Using the thermodynamic parameters shown in Supplementary Table 3, we calculated $V_B$, and then obtained $V_S$ and $\sigma$ from the calculated $V_B$ and measured $V_P$.

**Hugoniot velocity and density measurements on iron.** The compressional wave velocity $V_P$ of iron along the Hugoniot measured by Brown and McQueen[20] is often used to assess the effect of temperature on $V_P$ by comparing them with static data at room temperature. Figure 1a shows our experimental $V_P$ for iron at simultaneously high pressures and temperatures in the range of 56.8~234.0 GPa and 1073~5417 K, respectively. The data show a linear velocity-density relationship, $V_P = -2.89(\pm0.10) + 1.09(\pm0.02) \rho$, up to 180 GPa (corresponding to 11.77 g/cm³) for solid iron. At the same density, our measured $V_P$ values are 1.7~5.3% larger than previous data[20] by shock compression but are consistent with subsequent measurements by improved techniques[34]. The typical uncertainties in velocity measurements are 1.7~3.1% (Supplementary Tables 1 and 2). The calculated temperatures of our Hugoniot data range from 1073 K at 56.8 GPa to 3919 K at 180.1 GPa. However, the

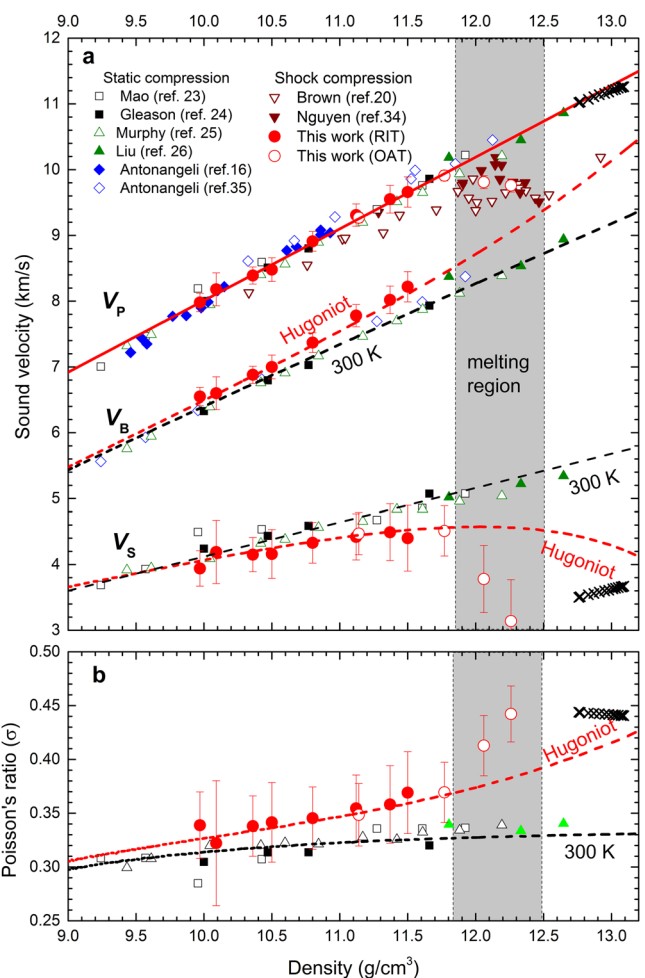

**Fig. 1 Sound velocities and Poisson's ratio of Fe as a function of density.**
**a** The measured $V_P$ along Hugoniot by the RIT (red solid circles) and the OAT (red open circles) are compared with previous shockwave data[20,34]. The solid red line represents the linear fit to the $V_P$ measurements for solid Fe. The static $V_P$ data by NRIXS[23–26] and IXS[16,35] are also plotted for comparison. The measured $V_B$ along Hugoniot by the RIT and the derived $V_S$ from this study are shown by red circles, compared with NRIXS data with a natural $^{57}$Fe isotope concentration at 300 K[23–26]. The black dashed line for $V_B$ is calculated from the equation of state at 300 K, and the red dashed $V_B$-line is the calculated $V_B$ from Hugoniot equation of state. The black and red dashed lines for $V_S$ represent the calculated results from $V_P$ and $V_B$, at 300 K and along Hugoniot, respectively. **b** The derived Poisson's ratios σ from this study are compared with the room-temperature NRIXS data. All measurements are compared with the observed values (black crosses) from PREM[14]. The boundaries of the melting region (shaded area) were determined according to the discontinuities of the sound velocity.

measured $V_P$ data along the Hugoniot are almost identical to the recent static data for hexagonal close-packed (hcp) iron at 300 K by IXS[35] and the corrected data (refer to the method[26]) by NRIXS[23–26] for iron with a natural $^{57}$Fe isotope concentration. The measured $V_P$ data are also consistent with the IXS data up to 1110 K obtained by stable external heating[16]. These results demonstrate that Birch's law still holds for the $V_P$ of iron at high temperature.

Our new Hugoniot density-pressure measurements are consistent with previous shock compression data[36] (Fig. 2). Using Hugoniot density-pressure data[36] and static 300 K isotherm[4], we calculate $V_P$ as a function of pressure constrained by Birch's law. Figure 2 shows consistent $V_P$ results at room temperature and along Hugoniot between the measurements[23–26] and calculations.

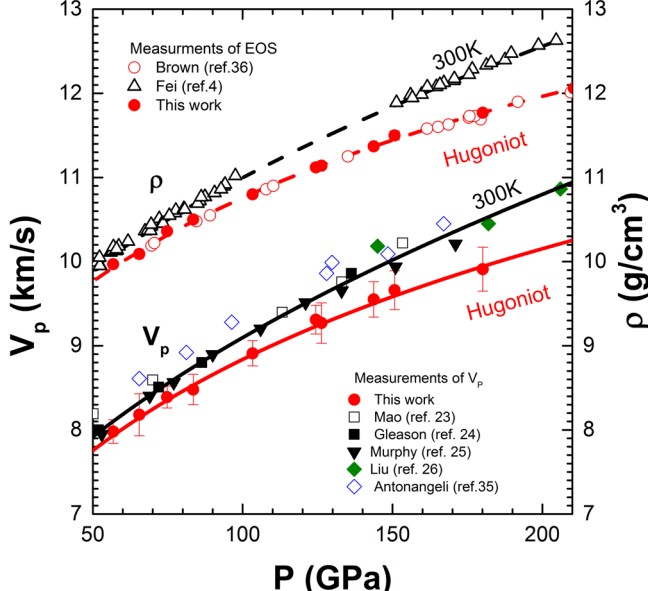

**Fig. 2 The density and compressional wave velocity ($V_P$) of Fe as a function of pressure along the Hugoniot and 300-K isotherms.** The Hugoniot densities of Fe[36] are compared with new Hugoniot data of this study and static data at 300 K[4], along with the fitted results represented by the dashed lines. The red solid circles represent the measured $V_P$ under shock compression (this work), compared with the static $V_P$ data[23–26,35]. The black and red curves represent the calculated $V_P$ as a function of pressure along 300 K and shock temperature, using Birch's law and equations of state of Fe.

Due to the quasi-harmonic and anharmonic effect at high temperature, both ρ and $V_P$ of iron under shock compression gradually deviate from the static room-temperature data at same pressure. But the linear relationship between $V_P$ and ρ does not change, indicating the anharmonic effect does not play a significant role in Birch's law. The linear density-velocity relationship defined by Birch's law would allow a reliable extrapolation of $V_P$ to core conditions for comparison with seismic models such as PREM[14].

Figure 1a also shows the direct measurements of $V_B$ from the RIT along with the calculated $V_B$ from equation of state. The $V_B$ obtained by both methods are in an excellent agreement. Static data[23–26] showed that the $V_B$ of pure iron increases linearly with density at 300 K. The shockwave data show a systematic deviation from the room-temperature data due to the temperature effect. From the measurements of $V_P$ and $V_B$, we calculate $V_S$ at the measured densities (Fig. 1a). At room temperature, $V_S$ linearly increases with density, expressed as $V_S = -1.08(\pm 0.03) + 0.52(\pm 0.03)\rho$. The $V_S$ along Hugoniot are systematically lower than the room-temperature values, showing shear softening with increasing temperature. The $V_S$ values drop significantly at 12.06 and 12.26 g/cm$^3$, indicating initiation of melting (Fig. 1a). Within the solid region, the effect of temperature on $V_S$ is dependent on density and temperature (Fig. 3a). The rate of change of $V_S$ in temperature, $|(\Delta V_S/\Delta T)_V|$, decreases with density by an exponential function and increases with temperature at a constant density. At 6000 K and 13.04 g/cm$^3$, relevant to inner core condition, $V_S$ of iron decreases at a rate of $|(\Delta V_S/\Delta T)_V| \approx 0.12$ ms$^{-1}$K$^{-1}$, much less than a calculated value of 0.48 ms$^{-1}$K$^{-1}$ by simulations[29]. The weak temperature dependence of $V_S$ makes more difficult to match the observed low $V_S$ in the inner core by the high-temperature effect alone.

Figure 1b shows the calculated Poisson's ratio σ that increases with density along the Hugoniot. The elevated σ values along the

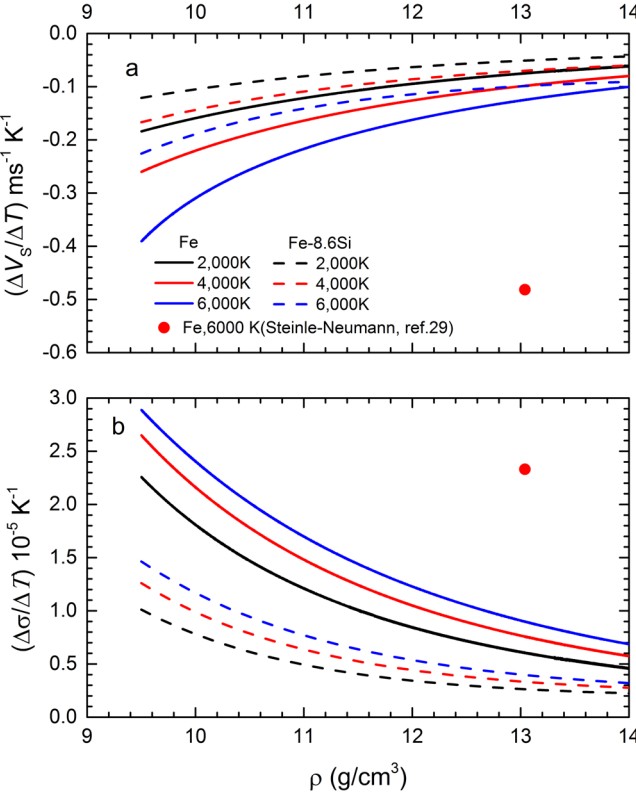

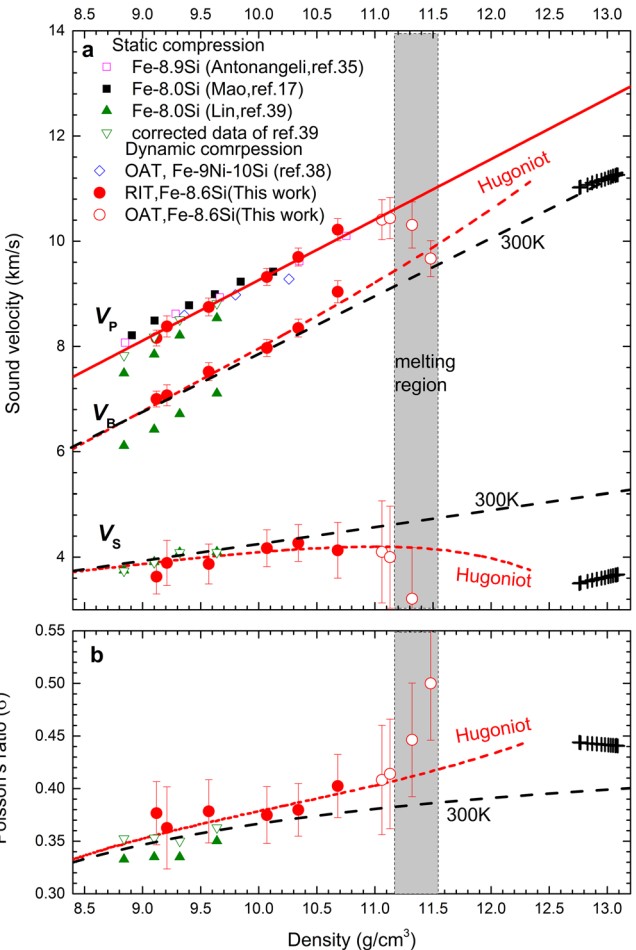

**Fig. 3 The rate of change of $V_S$ and $\sigma$ in temperature as a function of density at different temperatures. a** The calculated $\Delta V_S/\Delta T$ increase with density along 2000K (black line), 4000 K (red line) and 6000 K (blue line) isotherms. **b** The calculated $\Delta\sigma/\Delta T$ decrease with density along 2000K (black line), 4000 K (red line) and 6000 K (blue line) isotherms. The solid and dashed lines represent the values of Fe and Fe-8.6Si, respectively, compared with the *ab* initio calculations (red solid circle)[29].

Hugoniot relative to the 300 K isotherm indicate a positive effect by high temperature (Fig. 1b). Figure 3b shows the effect of temperature and density on $\sigma$. The $(\Delta\sigma/\Delta T)_V$ for iron can be expressed as an exponential function of density, with a positive temperature correlation. At 6000 K and 13.04 g/cm$^3$, $\sigma$ increases at a rate of $(\Delta\sigma/\Delta T)_V = 0.90 \times 10^{-5}$ K$^{-1}$, less than a predicted value of $2.33 \times 10^{-5}$ K$^{-1}$ by simulations[29].

Our new Hugoniot velocity measurements on pure iron below melting provide a direct evaluation of the temperature effect on the shear wave velocity and the Poisson's ratio. We also observed melting at a pressure between 180 and 210 GPa, indicated by a large drop in $V_P$ and $V_S$ and a fast rise in $\sigma$. The onset of melting of iron under shock compression has been an intense debate. The data by Brown and McQueen[20] showed two discontinuities in sound velocity at 200 GPa and 243 GPa, respectively, whose explanation has been controversial. New in-situ X-ray diffraction measurements by static compression have ruled out possible solid phase transition below melting[37]. Additional shock experiments by Nguyen and Holmes[34] showed a single discontinuity in $V_P$ of iron, indicating onset of melting at ~225 GPa. Our $V_P$ measurements indicate that experiments at 210 GPa and 234 GPa are in the melting region.

**Hugoniot velocity measurements on iron-silicon alloy.** In order to evaluate the effect of Si on the shear wave velocity, we also measured the sound velocities of Fe-8.6Si alloy by shock compression, using the same methodology described for the study of pure iron. Figure 4 shows the measurements of $V_P$ and $V_B$ for Fe-

**Fig. 4 Sound velocities and Poisson's ratio of Fe-8.6Si alloy as a function of density. a** The measured $V_P$ along Hugoniot by the RIT (red solid circles) and the OAT (red open circles) are compared with shockwave data for Fe-9Ni-10Si[38]. The solid red line represents the linear fit to the $V_P$ measurements for solid Fe-8.6Si alloy. The static $V_P$ data of Fe-8Si by NRIXS[39] and of Fe-8.9Si[35], Fe-8Si[17] by IXS are also plotted for comparison. The measures $V_B$ along Hugoniot by the RIT and the derived $V_S$ from this study are shown by red circles, compared with NRIXS data at 300 K[39]. The olive open inverted triangles are the corrected values based on the equation of state for Fe-8.6 Si[40] and the measured $V_D$ by NRIXS[39]. The black dashed line for $V_B$ is calculated from the equation of state at 300 K, and the red dashed $V_B$-line is the calculated $V_B$ from Hugoniot equation of state. The black and red dashed lines for $V_S$ represent the calculated results from $V_P$ and $V_B$, at 300 K and along Hugoniot, respectively. **b** The derived Poisson's ratios $\sigma$ from this study are compared with the room-temperature NRIXS data. All measurements are compared with the observed values (black crosses) from PREM[14]. The boundaries of the melting region (shaded area) were determined according to the discontinuities of the sound velocity.

8.6Si up to 162 GPa (corresponding to 10.68 g/cm$^3$) by the RIT (Supplementary Table 1). The $V_P$ measurements were extended to higher pressures between 208 and 239 GPa, with the OAT. We observed the onset of melting at 239 GPa based on the velocity drop. The estimated melting temperature is slightly lower than that of Fe-9Ni-10Si[38], which could be caused by uncertainties in the calculated Hugoniot temperature and the estimated porosity (Supplementary Fig. 6).

Similar to the results of Fe, the measured $V_P$ of the Fe–Si alloy also follows Birch's law, expressed by $V_P = -2.24(\pm0.10) + 1.15(\pm0.03)\rho$. The results are in agreements with shockwave data on solid Fe-9 wt% Ni-10 wt%Si alloy[38]. They are also comparable to the static room-

temperature data for hcp Fe-9 wt% Si[35] and Fe-8 wt% Si[17] by IXS, showing no resolvable difference between the room-temperature and Hugoniot data. Early $V_P$ measurements for Fe-8 wt% Si by NRIXS[39] showed systematically lower values compared to all other studies. The $V_P$ and $V_S$ by NRIXS were determined according to the calculated $V_B$ inferred from the equation of state and measurements of the Debye sound velocity ($V_D$) via $3V_D^{-3} = V_P^{-3} + 2V_S^{-3}$. Thus, $V_P$ is sensitive to the difference in the calculated $V_B$. At room temperature, the calculated $V_B$ of Fe-8.0Si[39] is systematically smaller than our calculations based on the equation of state of Fe-8.6Si[40] (Fig. 4a). Combining the $V_D$ measured by NRIXS[39] with the $V_B$ of Fe-8.6Si obtained in this work, the corrected $V_P$ of Fe-8.0 wt% Si is in excellent agreement with other static data by IXS and our Hugoniot data (Fig. 4a). The combined dataset shows no temperature effect on Birch's law.

Both datasets for pure Fe and Fe-8.6Si alloy demonstrated the validity of Birch's law. Using the linear $V_P$-$\rho$ relationship defined by Birch's law, we extrapolated $V_P$ of Fe-8.6Si to inner core conditions. The extrapolated value is consistent with the *ab initio* results for Fe-6.7Si between 0~4000 K[31], and the calculated $V_P$ for Fe-6.6 Si at 5500 K[30], but it is about 13.5% larger than the seismically observed value.

Our derived $V_S$ values from Hugoniot $V_P$ and $V_B$ measurements are smaller than those at room temperature, derived from NRIXS, showing temperature-induced softening. It is noted that the $V_S$ derived from NRIXS is insensitive to $V_B$ data, thus, the corrected $V_S$ values are indistinguishable from the original data of Fe-8.0 wt% Si[39]. The room-temperature $V_S$ is linearly related to the density, $V_S = 1.05(\pm0.04) + 0.32(\pm0.03)\rho$. The extrapolated data to inner-core density are much smaller than the simulated values at 0 K[30,31]. Under shock compression, $V_S$ gradually decreases (Fig. 4a) and $\sigma$ gradually increases (Fig. 4b) because of the effect of temperature. Similar to the results for pure iron discussed above, the magnitudes of $|(\Delta V_S/\Delta T)_V|$ and $(\Delta\sigma/\Delta T)_V$ for Fe-8.6Si also decrease with density by an exponential function and increase with temperature (Fig. 3). At 6000 K and $\rho = 13.04$ g/cm³, $V_S$ decreases at a rate of $(\Delta V_S/\Delta T)_V = -0.10$ ms$^{-1}$ K$^{-1}$, and $\sigma$ increases at a rate of $(\Delta\sigma/\Delta T)_V = 0.4 \times 10^{-5}$ K$^{-1}$. These rates are generally smaller than that of pure iron at the same conditions.

The above analysis of $V_P$, $V_S$ and $\sigma$ for Fe-8.6Si alloy is assumed no phase transition along the Hugoniot. Based on the calculated shock temperature of Fe-8.6Si[40] and the phase relations in Fe-Si alloys under static compression[41,42], it is possible that the shocked hcp Fe-8.6Si phase would decompose into a mixture of Si-poor hcp and Si-rich B2 (CsCl-type) phases at pressures above 145 GPa. However, there is no detectable change in the trends of $V_P$, $V_S$ and $\sigma$ above 145 GPa and 2400 K. This implies that either the decomposition process is kinetically prohibited because of the extremely short shock duration, or the physical properties of the decomposed mixture follow the ideal mixing rule[43]. Neither of these scenarios would affect our analysis because the equation of state and the compressional wave velocity for Fe–Si system can be interpolated using an ideal mixing model.

Using the ideal linear mixing model[43], we interpolated the $\rho$ and $V_P$ for Fe-6 wt%Si, Fe-4.5 wt% Ni-3.7 wt%Si, and Fe-9 wt% Ni-2.3 wt%Si alloys based on the corresponding values of Fe-23 wt% Ni[44], Fe-8.6Si, and Fe. The measured densities of several Fe–Si alloys from static compression experiments are consistent with the calculated results based on our ideal mixing model (Supplementary Fig. 7). The interpolated $V_P$ are in a good agreement with the measured $V_P$ of Fe-9 wt%Ni-2.3 wt%Si[26] and Fe-6 wt%Si[22], and broadly consistent with those of Fe-4.5 wt% Ni-3.7 wt%Si[21] by IXS (Fig. 5). The measured $V_P$ of Fe-5 wt%Si by picosecond acoustics measurements (PAM)[45] showed systematically higher values than the interpolation. Figure 5 illustrates that $V_P$ of Fe–Si system increase with the Si content of the Fe–Si

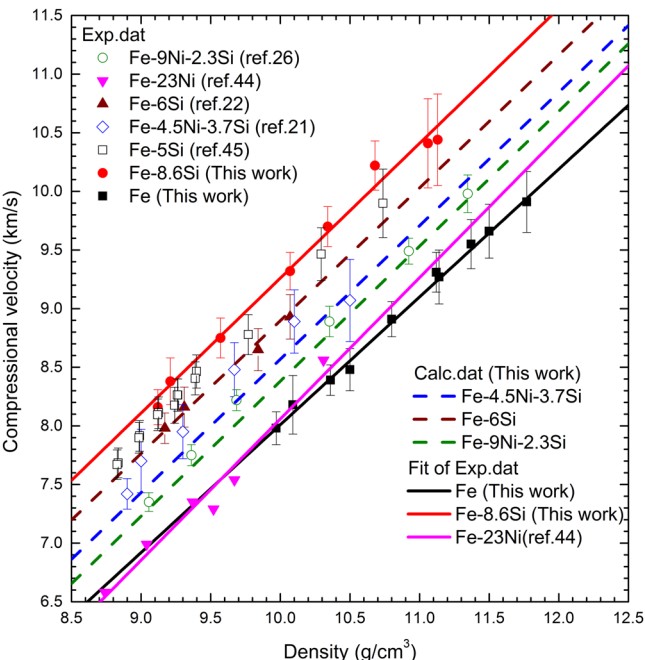

**Fig. 5 The relationship between density and compressional velocity for the Fe-Ni-Si system.** The black solid squares and red solid circles represent the new $V_P$ measurements for Fe and Fe-8.6Si, respectively, compared with static data for various intermediate compositions including Fe-6Si[22], Fe-4.5Ni-3.7Si[21], Fe-9Ni-2.3Si[26], and Fe-5Si[45]. The data for Fe-Ni alloy, Fe-23Ni[44], are also shown for comparison. The solid lines are the linear fits to the experimental data for the end members. The dashed lines represent the calculated $V_P$ for Fe-6Si, Fe-4.5Ni-3.7Si and Fe-9Ni-2.3Si (in weight percent) according to an ideal linear mixing model[43].

alloys. It should be noted that the slope of the sound velocity of Fe-Ni is larger than that of pure iron which could influence the extrapolation when Ni is considered as a core component. However, the data range is limited, and data are more scattered. Additional data for Fe-Ni alloy are required to incorporate Ni into the core models.

Compared with the $V_P$, the effect of Si on $V_S$ of the Fe–Si alloys is more complicated because $V_S$ is expected to decrease with temperature. Supplementary Fig. 8 shows that pure Fe has a stronger dependence of $V_S$ on density than the Fe-8.6Si alloy, leading to a higher $V_S$ for Fe than that of Fe-8.6Si at high density (>10.5 g/cm³). In contrast to $V_S$, $\sigma$ of Fe-8.6Si is almost parallel and ~16% larger than that of pure Fe at the same density (Supplementary Fig. 9). Therefore, the net effect is that an increase in the Si content in the alloy would decrease $V_S$ and increase $\sigma$ of the Fe–Si alloy under conditions of Earth's inner core.

## Discussion

We established a thermodynamic model (see Methods) with optimized parameters (Supplementary Table 3) that best describes the experimental data, and calculated the $\rho$, $V_P$, $V_S$ and $\sigma$ of Fe and Fe-8.6Si along an adiabatic temperature profile $T = T_{ICB} (\rho/\rho_{ICB})^{\gamma}$, with a Grüneisen parameter $\gamma$ of 1.5. The estimated temperature at the inner core boundary (ICB) $T_{ICB}$ ranges from 5200 K to 5700 K, and the calculations were based on a preferred value of 5440 K[3]. Figure 6 shows the calculated $\rho$, $V_P$, $V_S$ and $\sigma$ of Fe and Fe-8.6Si with uncertainties, compared with the PREM values in the inner core. The calculated values are listed in Supplementary Table 4. The uncertainties were assessed through propagation of errors associated with the parameters of equations

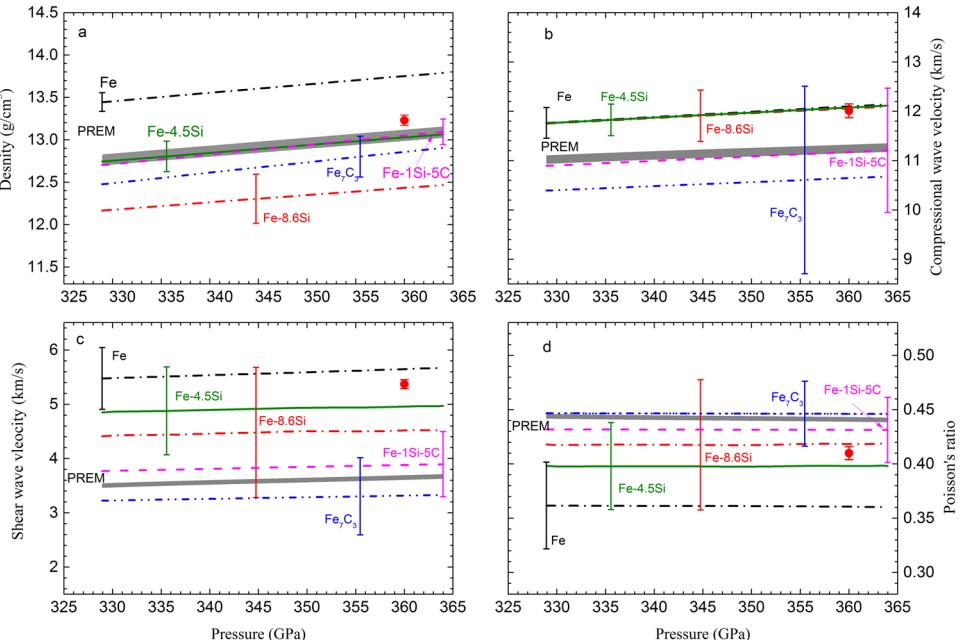

**Fig. 6 Comparison of densities, compressional wave velocities, shear wave velocities and Poisson's ratios of various core compositions with PREM values. a** The calculated densities of Fe (black dash-dot line), Fe-8.6Si (red dash-dot line), Fe-4.5 wt% Si (olive solid line), Fe$_7$C$_3$ (blue dash-dot-dot line) and Fe-1 wt% Si-5 wt% C mixture (magenta dash line) in the inner core assuming a temperature at inner core boundary of 5440 K, are compared to the PREM density profile. **b** The same comparison is shown for the compressional wave velocity. **c** The same comparison is shown for the shear wave velocity. **d** The same comparison is shown for the Poisson's ratio. The solid circles represent the simulated values for a model composition Fe$_{60}$Si$_2$C$_2$[31] at 360 GPa and 5500 K. The thick gray lines represent the PREM[14] for the inner core, with the line width representing 0.5% uncertainty in density, 0.5% in $V_P$ and 1.1% in $V_S$. The PREM uncertainty were derived by comparing different Earth's model[14,15]. Error bars are included for the calculated values. For the best-matched composition, Fe-1 wt% Si-5 wt% C, the error bars show 1.1 % errors in density, 11.4 % errors in $V_P$, 17.1 % errors in $V_S$ and 6.8% errors in $\sigma$.

of state, the Grüneisen parameters, and the parameters of Birch's law for $V_P$. The uncertainties in $\rho$, $V_P$, $V_S$ and $\sigma$ caused by the temperature uncertainty in the core were also assessed. A 10% uncertainty in $T_{ICB}$ contributes ~0.5% uncertainty in $\rho$ and $V_P$, ~1% uncertainty in $V_S$ and ~2% uncertainty in $\sigma$ for iron.

Throughout the Earth's inner core, the density of Fe and Fe-8.6Si are ~5.3% higher and ~4.6% lower than the PREM values, respectively. A Si content of 4.5 wt.% could explain the density deficit in the inner core (Fig. 6a). For comparison, we calculated the sound velocity and $\sigma$ for Fe-4.5Si alloy by interpolation. The $V_P$ values of Fe, Fe-4.5Si and Fe-8.6Si are almost coincident with each other, ~6.9% higher than the PREM (Fig. 6b). It is not surprising that the $V_P$ values of Fe–Si alloys are not sensitive to the Si content under inner core conditions because $V_P$ of Fe-Si alloy linearly increases with the Si content at constant density (Supplementary Fig. 10), while the density of Fe–Si alloy linearly decreases with the Si content at the same rate (Supplementary Fig. 11). The analyses show that incorporation of Si alone in the inner core cannot simultaneously satisfy both the density and compressional sound velocity requirements defined by the observations.

Under the conditions of the inner core, the $V_S$ of Fe–Si alloy decreases with the Si content. The $V_S$ of Fe-4.5Si is ~11 % less than that of Fe, but it is still ~38% larger than the PREM value even with a temperature-induced softening (Fig. 6c). The $\sigma$ of the Fe–Si system increases with the Si content under inner core's condition. Accounting for the temperature effect, the $\sigma$ value of pure Fe at 5440 K increases ~9 % compared with that at room temperature. As a net effect, the $\sigma$ value of Fe-4.5 Si is ~ 11% less than the PREM value (Fig. 6d).

The observed anomalously low $V_S$ and high $\sigma$ in the inner core were previously attributed to the temperature effect[29–31]. In this work, we provide the first estimate of $V_S$ and $\sigma$ from direct

measurements of both $V_P$ and $V_B$ under simultaneous high temperature and pressure, relevant to the core conditions, and find that neither the temperature nor silicon is the main factor that would explain the observed low $V_S$ and high $\sigma$ in the inner core. Although we observed a considerable decrease in $V_S$ and a considerable increase in $\sigma$ during melting (Figs. 1 and 4), these drastic changes were caused by the effect in the solid-liquid mixing-phase region and not associated with pre-melting behavior. Here, we do not consider that the pre-melting behavior[30,31] is a viable mechanism to explain the low $V_S$ and high $\sigma$ in the inner core, because the observed low shear velocity is not limited to the top region of the inner core.

The mismatch in both $V_P$ and $V_S$ for Fe–Si alloys would rule out Si as the dominant light element in the solid inner core. Any additional light elements incorporated in the inner core would have to further decrease $V_P$ and $V_S$ to match the observed inner core values. Compared with the available sound velocity measurements of FeH$_x$[46], Fe$_3$S[47,48], Fe$_7$C$_3$[27], and Fe$_3$C[49], iron carbide Fe$_7$C$_3$ (8.4 wt% C) is the only iron alloy that reduces the $V_P$ and $V_S$ comparing with pure Fe. Phase relations in the Fe–C system support that a mixture of Fe$_7$C$_3$ and Fe could be a potential composition of the inner core if the C content is high enough[50,51]. Therefore, we explore possible composition space for an Fe–Si–C core.

Measurements of the sound velocities ($V_P$ and $V_S$) of Fe$_7$C$_3$ at room temperature[27] yielded $V_P = 2.16(\pm1.16) + 0.66(\pm0.12)\rho$ and $V_{S,300K} = 0.843(\pm0.439) + 0.242(\pm0.045)\rho$. The $V_S$ values at high temperature were calculated by $V_S = V_{S,300K} + (dV_S/dT)_\rho$ (T − 300). Because there is no measurement of the temperature-dependence of $V_S$ for Fe$_7$C$_3$, we assumed that Fe$_7$C$_3$ has the same rate of change of $V_S$ at different densities for a given temperature as pure Fe (Fig. 3). Our measurements indicate the upper bound of $(dV_S/dT)_v$ for pure Fe is about −0.124 ms$^{-1}$K$^{-1}$ at an inner-

core temperature of 6000 K. Using the equation-of-state parameters (Supplementary Table 3) and the velocity functions described above, we calculated $\rho$, $V_P$, $V_S$ and $\sigma$ for $Fe_7C_3$ under core conditions and then compared them to the observed values. Throughout the Earth's inner core, $\rho$, $V_P$ and $V_S$ for $Fe_7C_3$ are ~2.2%, ~5.7%, and ~7.7% less than the inner core observations, respectively, whereas the $\sigma$ value of $Fe_7C_3$ is in the range of PREM. Thus, a combination of Si and C in the inner core could match PREM in all four key parameters ($\rho$, $V_P$, $V_S$, and $\sigma$). Using Fe, Fe-8.6Si, and $Fe_7C_3$ as the endmembers, the composition of Fe-1Si-5C (1 wt.% Si and 5 wt.% C) would yield the best fit to the observations (Fig. 6). The derived composition represents a global minimum using the best available thermodynamic parameters of the endmembers, and its uncertainty shown in Fig. 6 is propagated from errors associated with the parameters.

Our model calculations were based on interpolation from data of endmembers Fe, Fe-8.6Si and $Fe_7C_3$, using ideal mixing model[43]. Among the measured physical properties, the density measurement has the highest precision. To examine whether the equations of state of the Fe–C-Si alloys follow the ideal mixing model, we calculated the density of $Fe_{93}C_4Si_3$ (corresponding to 0.9 wt.% C and 1.6 wt.% Si) using data from the endmembers, Fe[4], Fe-8.6Si, $Fe_7C_3$[52] and $Fe_3C$[53], and compared the calculated results with the density measurements of the same alloy composition[54]. The agreement between the calculated and measured densities (Supplementary Fig. 12) suggests that the ideal mixing model is a reasonable interpolation for the analysis. The effect of C on the density in the core can be effectively modeled from the measurements of iron carbides if the C substitution is not interstitial in the structure, which may have a different effect[55].

The Fe–Si–C core model has also been tested using *ab* initio molecular dynamics calculations. Li et al.[31] calculated the density and velocities of hcp-FeSiC alloys for various composition combinations and found that the properties of a hcp-$Fe_{60}Si_2C_2$ (corresponding to 1.6 wt% Si and 0.7 wt% C) could match the density and sound velocity of the inner core, but at relatively high core temperature. The simulated $V_P$, $V_S$ and $\sigma$ for $Fe_{60}Si_2C_2$ at 360 GPa and 5500 K would deviate from the PREM values (Fig. 6).

Adding C into the inner core is merely based on the required match of physical properties between core alloys and the inner core. The incorporation of C into the solid inner core would have consequence for the chemistry of the core. Based on cosmochemical and geochemical consideration, the abundance of the carbon in the Earth's core ranges from ~0.2 to ~1 wt%[5,13]. Presence of ~5 wt.% C in the inner core would require that most of carbon entered the solid inner core during core solidification, with less than 0.8 wt.% C in the liquid outer core. Such a distribution of C between the inner and outer cores requires significant modification of melting relations in the Fe–C system at the ICB pressure, namely shifting the eutectic C composition toward the Fe endmember to keep the bulk C content of the core at the iron carbide + liquid region. This apparently contradicts with recent result on the evolution of the eutectic composition in the Fe–C system as a function of pressure, which did not detect significant change of the eutectic C content up to 260 GPa[51]. Similarly, a silicon-poor inner core requires to reconcile with seemingly robust conclusion of high Si content in the core from accretion and differentiation models[6,7]. The solution to this inner core paradox between the mineral physics models and geochemical and petrological constraints might lie in our understanding of melting relations in a multi-component iron alloy system at the ICB conditions. Therefore, further experiments should investigate the partitioning of C and Si between the solid and liquid iron at the inner core boundary, as well as the sound velocity of Fe–C–Si alloy under conditions relevant to the Earth's inner core.

## Methods

We used two-stage light gas guns at the Institute of Fluid Physics of China Academy of Engineering Physics and at the High Pressure Physics and Novel Materials Research Center in Wuhan University of Technology to measure the sound velocities of the Fe and Fe-8.6Si alloys. The discs of iron were from Trillion Metals Co., Ltd, with a chemical purity of 99.97(±0.05) wt% and an average density of 7.859(±0.007) g/cm³. The Fe–Si alloy powder with an average grain size of ~10 μm was from Goodfellow Co., Ltd., and was sintered in a large-volume cubic multi-anvil apparatus at 5 GPa and 1200 °C. The average bulk immersion density of the sintered sample is 7.386(±0.021) g/cm³. Electron microprobe analysis of the sintered discs showed uniform distribution of Si with an average Si content of 8.61(±0.02) wt%. The preparation of the Fe-8.6Si sample was described in our previous work[40].

The sound velocities of pure Fe and Fe-8.6Si alloy were measured using two complementary methods, the reverse-impact technique (RIT)[32,33] and optical analyzer technique (OAT)[20,33]. For the RIT, a 12-mm sample disc was used as a flyer to directly impact a LiF single-crystal window coated with a 3-μm aluminum film. An 8-μm aluminum foil was mounted in front of the film with epoxy (Supplementary Fig. 1a). Supplementary Fig. 1b shows the Lagrangian distance-time diagram for the reverse-impact experiments. The particle velocity history $u(t)$ at the film/LiF window interface was measured by the displacement interferometer system for any reflector (DISAR), as shown in Supplementary Fig. 2. When the flyer with thickness $h$ impacted the window at time $t_0$, indicated by a sharp increase in particle velocity, a shock wave with velocity $D_S$ and an elastic precursor wave with velocity $D_e$ were produced in the flyer. If $D_S$ is less than $D_e$, the elastic precursor wave $D_e$ first reaches the rear surface of flyer, then is reflected as a rarefaction wave with velocity $V_{P,0}$ (compressional velocity under ambient conditions), and finally meets the oncoming shock wave $D_S$ at position $h_1$[32].

$$h_1 = h \frac{D_s}{D_e} \left( \frac{D_e + V_{P,0}}{D_s + V_{P,0}} \right) \quad (1)$$

Then, $D_S$ is reflected as a rarefaction wave transported with velocity $V_{P,L}$ in the flyer and reaches the Al foil/LiF interface at time $t_1$, indicating a decrease in particle velocity (Supplementary Fig. 3). The compressional velocity in the Lagrangian coordinates is determined by

$$V_{P,L} = \frac{h_1}{(t_1 - t_0) - h_1/D_s} \quad (2)$$

If the plastic shock velocity $D_S$ is equal to or greater than the elastic precursor wave velocity at the Hugoniot elastic limit (HEL), only a single shock wave is observed, which is known as the "overdriven" condition. The overdriven conditions for Fe and Fe-8.6Si are expected to occur at stresses of 61(±2) GPa and 49(±2) GPa, respectively, based on their compressional velocities[56] under ambient condition. When the pressure is above the overdriven condition, position $h_1$ in Eqs. (1) and (2) is replaced with the flyer thickness $h$. Multiplying $V_{P,L}$ with $\rho_0/\rho$ (the ratio of the initial and compressed densities), we obtain the compressional wave velocity $V_P$ in Eulerian coordinates. The main sources of the uncertainty in the measured $V_P$ are from the uncertainties in the thickness $h$ of the sample, the shock wave $D_S$, the arrival time of the rarefaction wave $t_1$, and the initial and compressed densities. The errors in the measured $V_P$ for Fe and Fe-8.6Si range from 1.5~3% (Supplementary Table 1).

When the elastic-plastic transition (EPT) point at time $t_2$ during unloading is determined, the bulk velocities $V_B$ at released pressure can also be obtained by replacing $t_1$ with $t_2$ in Eq. (2). However, the EPT point at $t_2$ is very subtle in the particle velocity history record and difficult to determine precisely. To obtain $V_B$ precisely, we take the derivative of particle velocity $u(t)$ with respect to time (d$u$/d$t$). The d$u$/d$t$ data have a minimum value due to EPT which is considered as the inflexion of particle velocity history, $u(t)$. Supplementary Fig. 3a, b shows the particle velocity and its derivative as a function of time for experiment 170314. Supplementary Fig. 3c shows the Lagrangian sound velocity as a function of the particle velocity during unloading. During plastic unloading, the Lagrangian sound velocity increases linearly with the particle velocity. Extrapolating the linear relation to the particle velocity just before unloading yields a Lagrangian bulk velocity $V_{B,L}$ under compression. Multiplying $V_{B,L}$ by $\rho_0/\rho$, we obtain the bulk velocity $V_B$ in Eulerian coordinates. The main sources of the uncertainty in $V_B$ are the uncertainties in $h$, $D_S$, $t_2$ and the fitted parameters for the Lagrangian sound velocity during plastic unloading. The errors in the measured $V_B$ range from 1.9~3.8% (Supplementary Table 1).

From the direct measurements of $V_P$ and $V_B$, we obtained the shear wave velocity via $V_S = [3(V_P^2 - V_B^2)/4]^{1/2}$ and Poisson's ratio via $\sigma = 0.5(V_P^2 - 2V_S^2)/(V_P^2 - V_S^2)$, listed in Supplementary Table 1. Compared with $V_P$ and $V_B$, the propagated errors in $V_S$ and $\sigma$ are considerably large, ranging from 6~13% and 8~18%, respectively. Nevertheless, the derived $V_S$ and $\sigma$ along Hugoniot provide a direct evaluation of the temperature effect on shear wave velocity and Poisson's ratio when compared with static measurements at room temperature.

We also performed $V_P$ measurements on pure Fe and Fe-8.6Si alloy at higher pressure using the optical analyzer technique (OAT). Supplementary Fig. 4 illustrates the experimental configuration of a flyer (Fe or Ta) impacting the sample with different thicknesses and the Lagrangian distance-time diagram showing the flyer and sample interaction. Supplementary Fig. 5a–c show three particle velocity history records at the interface between the Ta foil and LiF window for the

experiment 180102 with three different sample thicknesses. A linear fit of the measured time interval as a function of the sample thickness provides the catch-up thickness (Supplementary Fig. 5d). The details of the sound velocity determination from the measured catch-up thickness of the sample and the known flyer properties were discussed by Huang et al[33]. The main sources of the uncertainty are from the uncertainties in the thickness of the sample, the shock velocities in the flyer and sample, and the compressional wave velocity in the flyer if its material is different from that of the sample. The uncertainty in $V_P$ measured by OAT ranges from 1~4% (Supplementary Table 2).

We also calculated the bulk sound velocities $V_B$ from the Hugoniot equation of state for solid Fe and Fe-8.6Si alloy based on following thermodynamic model.

$$V_B^2 = -V^2 \left(\frac{dP}{dV}\right)_{V,T} + \gamma_{eff}^2 C_V T \tag{3}$$

The value $(dP/dV)_{V,T}$ was obtained from the Grüneisen equation of state.

$$P(V,T) = P_{T_0}(V) + \int_{T_0}^{T} \frac{\gamma_{eff}}{V} C_V dT \tag{4}$$

where $\gamma_{eff}$, the effective Grüneisen parameter, includes the lattice and electrons contributions.

$$\gamma_{eff} = \frac{C_{V,l}\gamma_l + C_{V,e}\gamma_e}{C_{V,l} + C_{V,e}} \tag{5}$$

The lattice Grüneisen parameter is defined as $\gamma_l = \gamma_0 (\rho_0/\rho)^q$. $\gamma_e$ is the electrons Grüneisen parameter. $C_{V,l} = 3 R/\mu$ and $C_{V,e} = \beta_0 (\rho_0/\rho)^\kappa T$ are the specific heat contributed by the lattice and electrons, respectively. The parameters for Fe, Fe-8.6Si and Fe$_7$C$_3$ were listed in Supplementary Table 3. $P_{T_0}(V)$ is the equation of state at 300 K, which was calculated from Hugoniot pressure $P_H (V,T_H)$ by

$$P_{T_0}(V) = P_H(V, T_H) - \int_{T_0}^{T_H} \frac{\gamma_{eff}}{V} C_V dT \tag{6}$$

The shock temperature $T_H$ were calculated from the thermodynamic relation[20].

$$dT = \left[-T\frac{\gamma_{eff}}{V} + \frac{P_H + (V_0 - V)\frac{dP_H}{dV}}{2C_V}\right] dV \tag{7}$$

Using the third-order Birch–Murnaghan equation of state, the fitted parameters of isothermal bulk modulus $K_0$ and its pressure derivative $K_0'$ are given in Supplementary Table 3.

The calculated bulk sound velocity from the Hugoniot equation of state agrees well with the experimental measurements (Figs. 1 and 4). The main sources of uncertainty in the model calculations of $V_B$ are uncertainties in the Hugoniot parameters ($C_0$, $\lambda$), the lattice contributions to the Grüneisen parameter, and the electronic and anharmonic contributions to the specific heat. The thermodynamic parameters and their errors for Fe and Fe-8.6Si are listed in Supplementary Table 3. The uncertainties in the Hugoniot parameters directly contribute to the uncertainties in the derived equation of state, having a large influence on the propagated errors in $V_B$. Along Hugoniot, the propagated errors in the calculated $V_B$ for Fe and Fe-8.6Si are approximately 2% and 5%, respectively, which are smaller than the errors in the calculated $V_S$ in the ranges of 7~20% and 23~30%, respectively. The errors in $\sigma$ for Fe and Fe-8.6Si are 6~8% and 12%, respectively.

Under the conditions of Earth's inner core, the density and sound velocity in the Fe–Si–C system are estimated based on an ideal mixing model

$$\rho = 1 \Big/ \sum_i \frac{W_i}{\rho_i} \tag{8}$$

and

$$V_{P(or\ S)} = 1 \Big/ \left(\rho \sum_i \frac{W_i}{\rho_i V_{P(or\ S),i}}\right) \tag{9}$$

where $W_i$ is the weight percent, $V_{P(or\ S),i}$ and $\rho_i$ represent the sound velocity and density of the endmember (Fe, Fe-8.6Si or Fe$_7$C$_3$) at the same pressure and temperature. For Fe and Fe-8.6Si, $\rho$ and $V_B$ were calculated with Eqs. (3) and (4), and $V_P$ data were calculated using Birch's law. The $V_S$ and $\sigma$ were then calculated from $\rho$, $V_B$ and $V_P$. Because of no direct shock wave measurement for Fe$_7$C$_3$, the thermodynamic parameters for Fe$_7$C$_3$ at high pressure and temperature have not been constrained. We therefore estimated the $V_S$ of Fe$_7$C$_3$ using the data at 300 K with an assumed $(dV_S/dT)_V$ value. The errors of the density and sound velocity of the mixture were calculated based on the following equations.

$$\delta\rho = \left[\sum\left(\frac{\partial\rho}{\partial\rho_i}\delta\rho_i\right)^2\right]^{1/2} \tag{10}$$

and

$$\delta V_{P(or\ S)} = \left[\sum\left(\frac{\partial V_{P(or\ S)}}{\partial\rho_i}\delta\rho_i\right)^2 + \sum\left(\frac{\partial V_{P(or\ S)}}{\partial V_{P(or\ S),i}}\delta V_{P(or\ S),i}\right)^2\right]^{1/2} \tag{11}$$

where the subscript P(or S) indicates the compressional or shear velocity. The subscript $i$ represents the $i$th endmember (Fe, Fe-8.6Si or Fe$_7$C$_3$) at the same

pressure and temperature. $\delta\rho_i$ and $\delta V_{P(or\ S),i}$ represent the errors of density and sound velocity of the $i$th endmember.

## Data availability
All data generated or analyzed during this study are included in this published article and its Supplementary Information files.

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

## Acknowledgements

We thank Q. Shen (the Sate Key Lab of Advanced Technology for Materials Synthesis), M. Yang and C. Shen (the Materials Research and Testing Center) for sample preparation, and R. Tao for chemical analysis of the samples. This work was supported by the National Science Foundation of China (grant 41874103, 51932006), and the Fundamental Research funds for the Central Universities (grants WUT 14050). Additional support for this collaborative research was provided by the Carnegie Institution for Science, and National Science Foundation grants (EAR-1619868) to YF.

## Author contributions

H.H. and Y.F. designed the project. H.H., L.F., X.L., F.X., Y.W., G.Y., C.L., Q.W., J.W., X.W., and L.C. participated the data collection and analysis. H.H and Y.F were responsible for data interpretation and wrote the manuscript. L.F., X.L., F.X., Y.W., G.Y., C.L., Q.W., J.W., X.W., and L.C. participated the discussion and provided comments on the manuscript.

## Competing interests

The authors declare no competing interests.
