## [Peer Review File · Nature Communications]

Inner core composition paradox revealed by shear wave velocities of Fe and Fe-Si alloyREVIEWER COMMENTS

Reviewer #1 (Remarks to the Author):

This manuscript is presenting new experimental determination of sound velocity and density of Fe and Fe-Si alloy using dynamic compression. The dataset is complemented with calculations of shear velocity and Poisson ratio under inner core conditions, to decipher the composition of the inner core.

The manuscript is well-written, with some small spelling mistakes. My main concern regarding the present manuscript is that the authors did not consider the recent publications around Fe-Si-C ternary system recently published in the last two years, which would be really important in their discussion. Overall, the dataset and the mineral physics conclusion are of high quality. I would therefore recommend the publications of the present manuscript after a deep update related to the recent literature.

Detailed comments

Introduction

Line 31 : "Recent ... models", I have the feeling that these models are not that recent ... Maybe a more detailed comment of the recent literature around metal/silicate partitioning would be required here to discuss the Si or O budget in the core, as well as the loss of volatile elements. This should be also put in perspective in the conclusion, as the present paper state a relatively high C content in the core.

Line 38-42: This part is a bit weird. First, the authors state that there are no data on Fe at core pressure. And the sentence after, they discuss the difference between seismological observables and pure Fe at core conditions. So there are data at core pressures! I would recommend to make a more extensive discussion of experimental (static or dynamic) and theoretical dataset, the latest allowing the comparison with observables.

Results

The fact that the Birch law is still valid is a relatively important message from the present article. However, there are experimental results using LH-DAC setup with a somehow different result, finding some temperature effect on the Birch law, in agreement with Brown data (Lin et al., 2005; Sakamaki et al., 2016). Similar discussion could be performed by comparison with recent data on hcp Fe-Si at high temperature (Sakairi et al., 2018). The authors should discuss the fact that static compression find a deviation with the Birch law.

The decrease of the VB induced by melting is relatively clear on the present dataset. What is the temperature of melting found for the Fe-Si alloy? Is it in agreement with static compression dataset? Could the author make supplementary figures presenting the Hugoniot in a P-T graph, and comparing with previous static and dynamic phase diagram results? Such comparison would be helpful, but not central to the present manuscript.

Discussion and conclusion

Line 156 : The authors discuss a composition of 3.6wt%Si that would fit the density, but not the other seismological observations. I would suggest to present this on the different graphs of Figure 3, instead of the 8.6 wtSi, that is not in agreement with any of the observables. With this figure, it is not possible to see the fact that Si could not satisfy the four presented observables.

Also, I am wondering what are the uncertainties on the observables? Could the author present them on the Figure 3?

Without comparing with the strongest effect due to premelting, how the present effect of T on VS could be compared with numerical simulations results? This should be commented also...

Line 163 : larger instead of "lager"

Line 188: ideal instead of "idea"

Regarding the effect of carbon and silicon on the phase diagram and elastic properties of hcp Fe, the authors should compare their results with the present list of recent bibliography :

(Edmund et al., 2019; Li et al., 2018; Miozzi et al., 2020; Pamato et al., 2020)

These different papers discuss the compatibility of an FeSiC ternary system as an inner core.

References:

Edmund, E., Antonangeli, D., Decremps, F., Miozzi, F., Morard, G., Boulard, E., Clark, A. N., Ayrinhac, S., Gauthier, M., Morand, M., & Mezouar, M. (2019). Velocity-Density Systematics of Fe-5wt%Si: Constraints on Si Content in the Earth's Inner Core. *Journal of Geophysical Research*:

Solid Earth, 124(4), 3436–3447. <https://doi.org/10.1029/2018JB016904>

Li, Y., Vocadlo, L., & Brodholt, J. P. (2018). The elastic properties of hcp -Fe alloys under the conditions of the Earth ' s inner core. *Earth Planet. Sc. Lett.*, 493, 118–127. <https://doi.org/10.1016/j.epsl.2018.04.013>

Lin, J.-F., Sturhahn, W., Zhao, J., Shen, G., Mao, H., & Hemley, R. J. (2005). Sound Velocities of Hot Dense Iron: Birch's Law Revisited. *Science*, 308(5730), 1892–1894. <https://doi.org/10.1126/science.1111724>

Miozzi, F., Morard, G., Antonangeli, D., Baron, M. A., Boccato, S., Pakhomova, A., Garbarino, G., Mezouar, M., & Fiquet, G. (2020). Eutectic melting of a ternary Fe-Si-C alloy up to 200 GPa and implications for the Earth's core. *Earth Planet. Sc. Lett.*, 544, 116382.

Pamato, M. G., Li, Y., Antonangeli, D., Miozzi, F., Morard, G., Wood, I. G., Vočadlo, L., Brodholt, J. P., & Mezouar, M. (2020). Equation of State of hcp Fe-C-Si Alloys and the Effect of C Incorporation Mechanism on the Density of hcp Fe Alloys at 300 K . *Journal of Geophysical Research: Solid Earth*, 125(12), 1–13. <https://doi.org/10.1029/2020jb020159>

Sakairi, T., Sakamaki, T., Ohtani, E., Fukui, H., Kamada, S., Tsutsui, S., Uchiyama, H., & Baron, A. Q. R. (2018). Sound velocity measurements of hcp Fe-Si alloy at high pressure and high temperature by inelastic X-ray scattering. *American Mineralogist*, 103, 85–90.

Sakamaki, T., Ohtani, E., Fukui, H., Kamada, S., Takahashi, S., Sakairi, T., Takahata, A., Sakai, T., Tsutsui, S., Ishikawa, D., Shiraishi, R., Seto, Y., Tsuchiya, T., & Baron, A. Q. R. (2016). Constraints on Earth's inner core composition inferred from measurements of the sound velocity of hcp-iron in extreme conditions. *Science Advances*, 2(2), e1500802–e1500802. <https://doi.org/10.1126/sciadv.1500802>

Reviewer #2 (Remarks to the Author):

This study consists of measurements of density, compressional velocity, and bulk sound speed on iron and an iron-silicon alloy from shock experiments. These values are used to calculate shear velocity and Poisson's ratio and demonstrate that an iron-silicon alloy cannot match all the measured values of these properties in the inner core. The study goes on to propose that an ideal mixture of their measured iron-silicon alloy and an iron-carbon alloy can match all the inner core values, despite this mixture being geochemically unlikely (thus the "paradox").

Examinations of iron alloys at core relevant conditions are fundamental to mineral physics. The experimental data in this study appear to be novel, well collected, and well analyzed. The conclusion that neither temperature nor Si alloying can account for the measured inner core properties is quite interesting. However, the extension of the analysis to carbon-bearing alloys is less well supported. No velocity data were collected on iron-carbon alloys, so it is unclear why C (and Fe₇C₃ in particular) was chosen above all other light element alloys. On top of this, there is insufficient discussion of important calculations throughout the paper, and the properties of the alloy mixture should be more carefully treated.

General comments

- Grammar and typos

This manuscript, while not by any means difficult to read, contains an unusual number of errors in grammar and word choice. The text should be carefully proofread to fix these mistakes and ensure there are not any similar typos in the numbers or figures.

- Quantity of supplemental information

There is a lot of data in the supplement, and it is repeatedly referenced in the main text. The supplementary information currently contains significant portions of the results/analysis, and some of it should be moved unless there is a journal-imposed restriction. In particular, Figures S6, S7, and S8 would be of interest to most readers (S7 and S8 can probably be combined).

- Insufficient description of Birch's Law

Birch's Law is invoked throughout the paper but is never explicitly defined or given a citation. Readers should be told what Birch's Law is, how you determined that your data are sufficiently linear to justify using it, and the significance of its applicability.

- Insufficient justification of ideal mixing

The analysis part of this study relies on the ideal mixing approximation holding for solid Fe alloys

at inner core conditions. Is this likely to be true? Are there applicable non-ideal mixing models?

- Introduction of C is confusing

The central “paradox” referenced in the title of this study is the difficulty in matching inner core physical properties with an Fe–Si alloy (which is predicted by geochemistry). This study says as much based on their experimental data, but then goes on to say that a C-bearing alloy (which was not part of their experimental dataset) could match all the geophysical parameters. Several things are unclear to me about this analysis:

o Why was C selected? Could the parameters be matched with H, S, O alloys?

o The values needed for V_p and V_s of Fe₇C₃ are not found in Table S3. Please be more explicit about how the calculation was performed, which values were used from references 16 and 33, and their uncertainties.

o Why choose an alloy (Fe₇C₃) with no high temperature velocity data available? To what degree does the imposed $(dV_S/dT)_P$ introduce uncertainty? Reference 33, for example, uses a wide range of $(dV_S/dT)_P$ rather than the fixed value used here.

o How was the “best fit” Fe–Si–C composition found? The error bars in Fig. 3 seem large enough that many compositions could have fit equally well.

Line comments

- 11: Change “experimental” to “experiments”
- 15: This line says that there have been no V_s or σ measurements but Figures 1 and 2 show preexisting data on Fe and Fe–Si. Please clarify.
- 24: Change “seemly” to “seemingly”
- 26: Final sentence of paragraph should be rewritten.
- 41: There should be parentheses around σ
- 45: You should clarify which quantities were measured versus calculated.
- 78: I assume “natural isotopic abundance” implies that you corrected for ⁵⁷Fe enrichment in NRIXS data? If so, say that explicitly, perhaps in the relevant figure captions.
- 81: Final sentence of paragraph should be rewritten.
- 102: These pressures are not, in fact, consistent with the cited study, which finds melting from 225-260 GPa.
- 105: Why was this Si content chosen?
- 153: A 6000 K ICB was an assumption even in the referenced study. Perhaps briefly justify this temperature choice.
- 154: The “M” in PREM already stands for “model”. Also, there have been recent reevaluations of inner core properties (e.g., 10.1126/science.aav2296 and references therein), is PREM still the best point of comparison for this study?
- 158: Perhaps similarity would be “not so surprising” but Fe and Fe-8.6Si appear identical. There are several studies which show at least a small effect of Si content on V_p (e.g., 10.1029/2003GL018405).
- 166: How much higher than that of Fe?
- 170: How is “near melting” different than “pre-melting”? Is there any melting outside of the shaded “melting region”?
- 185: Mention that this is ~8 wt% C.
- 198: This sentence just repeats Line 190.
- 201: The core C content is controversial, maybe this should be reframed as the required 4% inner core C requiring a certain bulk core or bulk Earth abundance.
- 202: You should cite a study of solid-liquid alloy C partitioning and comment on whether it is reasonable for the inner core to have all the C.
- 208: Be more specific about the types of data that could help explore this problem.
- Figure 1: Please clarify how the boundaries of the “melt region” were determined and the associated pressures. I recommend using a different color for the Nguyen data and adjusting the “melting region” bar so that it does not cover the axes and tick marks.
- Figure 2: Same as Figure 1.
- Figure 3: Plotting the Si-only best match (Fe-3.6Si) would make it clearer why another light element is required. I recommend changing the way the PREM line is plotted (to a continuous line instead of unevenly spaced symbols) and labeling PREM on the line instead of in the corner. The properties of Fe-1Si-4C were interpolated from two other alloys (Fe₇C₃ and Fe-8.6Si) so how is it possible that Fe-1Si-4C has the smallest uncertainty of the three?
- Figure S9: This figure appears to show a notable effect of Ni on Fe alloys which is not examined

in the main text. Would the lines in Figure 3 change if they were recalculated with a realistic Ni component?

Responses to the reviewers' comments:

Reviewer #1

This manuscript is presenting new experimental determination of sound velocity and density of Fe and Fe-Si alloy using dynamic compression. The dataset is complemented with calculations of shear velocity and Poisson ratio under inner core conditions, to decipher the composition of the inner core.

The manuscript is well –written, with some small spelling mistakes. My main concern regarding the present manuscript is that the authors did not consider the recent publications around Fe-Si-C ternary system recently published in the last two years, which would be really important in their discussion. Overall, the dataset and the mineral physics conclusion are of high quality. I would therefore recommend the publications of the present manuscript after a deep update related to the recent literature.

We thank reviewer for the comments and providing relevant references. We have added a discussion with the relevant references in the revision.

Detailed comments

Introduction

[1] Line 31 : “Recent ... models”, I have the feeling that these models are not that recent ... Maybe a more detailed comment of the recent literature around metal/silicate partitioning would be required here to discuss the Si or O budget in the core, as well as the loss of volatile elements. This should be also put in perspective in the conclusion, as the present paper state a relatively high C content in the core.

We have revised the introduction paragraph (line 27-33) to include background information and relevant references.

[2] Line 38-42: This part is a bit weird. First, the authors state that there are no data on Fe at core pressure. And the sentence after, they discuss the difference between seismological observables and pure Fe at core conditions. So there are data at core pressures! I would recommend to make a more extensive discussion of experimental (static or dynamic) and theoretical dataset, the latest allowing the comparison with observables.

We re-wrote this part of introduction in the revised manuscript and provided more extensive discussion of the available experimental (static and dynamic) and theoretical data (lines 34-72). The emphasis is on the sound velocity measurements, particularly regarding the lack of direct measurements of the shear sound velocity under simultaneous high pressure and temperature conditions.

[3] *The fact that the Birch law is still valid is a relatively important message from the present article. However, there are experimental results using LH-DAC setup with a somehow different result, finding some temperature effect on the Birch law, in agreement with Brown data (Lin et al., 2005; Sakamaki et al., 2016). Similar discussion could be performed by comparison with recent data on hcp Fe-Si at high temperature (Sakairi et al., 2018). The authors should discuss the fact that static compression find a deviation with the Birch law.*

As suggested, we have added a discussion about the temperature effect on the Birch's law (lines 40-49).

[4] *The decrease of the V_B induced by melting is relatively clear on the present dataset. What is the temperature of melting found for the Fe-Si alloy? Is it in agreement with static compression dataset? Could the author make supplementary figures presenting the Hugoniot in a P - T graph, and comparing with previous static and dynamic phase diagram results? Such comparison would be helpful, but not central to the present manuscript.*

Thank you for the suggestion. The decrease of V_P is related to melting. We observed the onset of melting at 239 GPa for Fe-8.6Si alloy based on the velocity drop. We added **Supplementary Figure 6** to show the estimated melting curve from this study. The result is compared with previous static and dynamic data. We also discuss the Hugoniot P - T path in relation to the phase stability in the Fe-Si system.

[5] *Line 156 : The authors discuss a composition of 3.6wt%Si that would fit the density, but not the other seismological observations. I would suggest to present this on the different graphs of Figure 3, instead of the 8.6 wtSi, that is not in agreement with any of the observables. With this figure, it is not possible to see the fact that Si could not satisfy the four presented observables.*

Thank you for the suggestion. We emphasized that the density fit to that of the inner core is also sensitive to the assumed temperature at the ICB. In **Figure 6** of the revision (Figure 3 in the original version), we showed the results for Fe-4.5wt% Si that would explain the density deficit in the inner core assuming an ICB temperature of 5440 K. However, its V_P and V_S are higher than the observations as illustrated in **Figure 6**.

[6] *Also, I am wondering what are the uncertainties on the observables? Could the author present them on the Figure 3?*

In the revision, we added the uncertainties in **Figure 6** (Figure 3 in the original version) and discussed the sources of uncertainties in the text.

[7] *Without comparing with the strongest effect due to premelting, how the present effect of T on V_S could be compared with numerical simulations results? This should be commented also...*

We did not consider that the pre-melting behavior is a viable mechanism to explain the low V_S and high σ in the inner core, because the observed low shear velocity is not limited to the top region of the inner core. The goal of this study is to quantify the effect of temperature on V_S for Fe alloy at relevant core temperatures. We added **Figure 3** in the revision to show $|(\Delta V_S/\Delta T)_T|$ as a function of density and temperature, compared with estimates from simulations.

[8] *Line 163 : larger instead of “lager”*
Corrected.

[9] *Line 188: ideal instead of “idea”*
Corrected.

[10] *Regarding the effect of carbon and silicon on the phase diagram and elastic properties of hcp Fe, the authors should compare their results with the present list of recent bibliography : (Edmund et al., 2019; Li et al., 2018; Miozzi et al., 2020; Pamato et al., 2020) These different papers discuss the compatibility of an FeSiC ternary system as an inner core.*

Thank you for the suggestion. We added **Supplementary Figure 12** to compare the compositional effect of the equation of state in the Fe-Si-C system. To examine whether the equations of state of the Fe-C-Si alloys follow the ideal mixing model, we calculated the density of $\text{Fe}_{93}\text{C}_4\text{Si}_3$ (corresponding to 0.9 wt.% C and 1.6 wt. % Si) using data from the endmembers, Fe, Fe-8.6Si, and Fe_7C_3 , and compared the calculated results with the density measurements of the same alloy composition. The agreement between the calculated and measured densities (**Supplementary Figure 12**) suggests that the ideal mixing model is a reasonable interpolation for the analysis.

Reviewer #2:

[1] This study consists of measurements of density, compressional velocity, and bulk sound speed on iron and an iron-silicon alloy from shock experiments. These values are used to calculate shear velocity and Poisson's ratio and demonstrate that an iron-silicon alloy cannot match all the measured values of these properties in the inner core. The study goes on to propose that an ideal mixture of their measured iron-silicon alloy and an iron-carbon alloy can match all the inner core values, despite this mixture being geochemically unlikely (thus the “paradox”).

Examinations of iron alloys at core relevant conditions are fundamental to mineral

physics. The experimental data in this study appear to be novel, well collected, and well analyzed. The conclusion that neither temperature nor Si alloying can account for the measured inner core properties is quite interesting. However, the extension of the analysis to carbon-bearing alloys is less well supported. No velocity data were collected on iron-carbon alloys, so it is unclear why C (and Fe₇C₃ in particular) was chosen above all other light element alloys. On top of this, there is insufficient discussion of important calculations throughout the paper, and the properties of the alloy mixture should be more carefully treated.

Thank you for the positive comments. In the discussion section, we added justification to introduce carbon to the system (lines 267-272). Compared with the available sound velocity measurements of FeH_x, Fe₃S, Fe₇C₃, and Fe₃C, iron carbide Fe₇C₃ (8.4wt% C) is the only iron alloy that reduces the V_P and V_S comparing with pure Fe. Phase relations in the Fe-C system support that a mixture of Fe₇C₃ and Fe could be a potential composition of the inner core if the C content is high enough. Therefore, we explore possible composition space for an Fe-Si-C core. Additional supplementary figures have been added to support the ideal mixing model for the analysis (see point [5] below).

[2] Grammar and typos

This manuscript, while not by any means difficult to read, contains an unusual number of errors in grammar and word choice. The text should be carefully proofread to fix these mistakes and ensure there are not any similar typos in the numbers or figures.

We have made an effort to correct the errors in grammar and typos. Thank you for the specific comments which are very helpful for the improvement of the manuscript.

[3] There is a lot of data in the supplement, and it is repeatedly referenced in the main text. The supplementary information currently contains significant portions of the results/analysis, and some of it should be moved unless there is a journal-imposed restriction. In particular, Figures S6, S7, and S8 would be of interest to most readers (S7 and S8 can probably be combined).

Thank you for the suggestion. We have moved three important figures to the main text, as suggested.

[4] Insufficient description of Birch's Law. Birch's Law is invoked throughout the paper but is never explicitly defined or given a citation. Readers should be told what Birch's Law is, how you determined that your data are sufficiently linear to justify using it, and the significance of its applicability.

We have added additional discussion about the Birch's law in the revision (Lines 40-49). We provided the fitted V_P - ρ linear equations with uncertainties in the text.

[5] • Insufficient justification of ideal mixing

The analysis part of this study relies on the ideal mixing approximation holding for solid Fe alloys at inner core conditions. Is this likely to be true? Are there applicable non-ideal mixing models?

We tested the ideal mixing model to interpolate densities of alloys using endmembers in the Fe-Fe_{8.6}Si system (**Supplementary Figure 7**) and the Fe-Si-C system (**Supplementary Figure 12**). We also demonstrated the interpolation for the velocity-density relations of alloys in the Fe-Ni-Si system, as shown in **Figure 5**.

[6] Introduction of C is confusing

The central “paradox” referenced in the title of this study is the difficulty in matching inner core physical properties with an Fe–Si alloy (which is predicted by geochemistry). This study says as much based on their experimental data, but then goes on to say that a C-bearing alloy (which was not part of their experimental dataset) could match all the geophysical parameters. Several things are unclear to me about this analysis: Why was C selected? Could the parameters be matched with H, S, O alloys?

As discussed above, we added justification to introduce carbon to the system (lines 267-272). C is chosen because iron carbide Fe₇C₃ (8.4wt% C) is the only iron alloy that reduces the V_P and V_S comparing with pure Fe, based on available data. We re-wrote the last paragraph (lines 302-319) to highlight the “paradox” in satisfying inner core physical properties with the constraints from geochemical and experimental petrological data.

[6a] The values needed for V_P and V_S of Fe₇C₃ are not found in Table S3. Please be more explicit about how the calculation was performed, which values were used from references 16 and 33, and their uncertainties.

In the revision, we added the method to calculate the sound velocity for Fe₇C₃ (lines 273 - 285). We also listed relevant EOS parameters for Fe₇C₃ in **Supplementary Table 3**.

[6b] Why choose an alloy (Fe₇C₃) with no high temperature velocity data available? To what degree does the imposed $(dV_S/dT)_P$ introduce uncertainty? Reference 33, for example, uses a wide range of $(dV_S/dT)_P$ rather than the fixed value used here.

There is no measurement of V_P and V_S for Fe₇C₃ at simultaneously high pressure and temperature so far. In the revision, we firstly estimated the density of Fe₇C₃ under the condition of the Earth’s inner core. Then we obtain the V_P and V_S at 300 K by linear extrapolation, similar to previous approach. Because of the lack of experimental constraint on dV_S/dT , it is a common practice to examine a range of values. Here, we made an assumption of the likely values based on our measurements on iron alloy. It provided at least a tight upper bound value.

[6c] How was the “best fit” Fe–Si–C composition found? The error bars in Fig. 3 seem large enough that many compositions could have fit equally well.

We calculate the density, V_p and V_s of Fe–Si–C system with different content of C and Si, and then find the best fit. The uncertainties in **Figure 6** propagated from errors associated with the parameters used in the equations of state, the Grüneisen parameters, and the parameters of Birch’s law for V_p . From the error bars in **Figure 6** (Figure 3 in the original version), it is clear there is a trade-off between Si and C contents. However, significant amounts of C need to be incorporated in order to meet the velocity requirement.

[7] line 11: Change “experimental” to “experiments”
Changed.

[8] line 15: This line says that there have been no V_s or σ measurements but Figures 1 and 2 show preexisting data on Fe and Fe–Si. Please clarify.

The V_s and σ data by static compression were measured at high pressure and room temperature. We re-worded the statement to emphasize the lack of V_s and σ data under simultaneous high P-T conditions.

[9] line 24: Change “seemly” to “seemingly”
Changed.

[10] line 26: Final sentence of paragraph should be rewritten.

We have re-written the last paragraph to reflect the conclusion of the manuscript.

[11] line 41: There should be parentheses around σ
Corrected.

[12] line 45: You should clarify which quantities were measured versus calculated.

In this study, V_p and V_B of Fe and Fe-8.6Si were directly measured using the reverse-impact technique. V_s and σ were calculated based on the measurements. For the optical analyzer technique, only V_p were directly measured, whereas V_B were calculated from the equation of state. We added a description in the **Method**.

[13] line 78: I assume “natural isotopic abundance” implies that you corrected for ^{57}Fe enrichment in NRIXS data? If so, say that explicitly, perhaps in the relevant figure captions.

It is correct that the natural isotopic abundance implied the corrected data for ^{57}Fe enrichment in NRIXS data, and we clarified the point in the revised figure captions.

[14] line 81: Final sentence of paragraph should be rewritten.
Changed.

[15] line 102: These pressures are not, in fact, consistent with the cited study, which finds melting from 225-260 GPa.

We have revised the sentences as “*The data by Brown and McQueen²⁰ showed two discontinuities in sound velocity at 200 GPa and 243 GPa, respectively, whose explanation has been controversial. New in-situ X-ray diffraction measurements by static compression have ruled out possible solid phase transition below melting³⁷. Additional shock experiments by Nguyen and Holmes³⁴ showed a single discontinuity in V_P of iron, indicating onset of melting at ~225 GPa. Our V_P measurements indicate experiments at 210 GPa and 234 GPa in the melting region.*”

[16] line 105: Why was this Si content chosen?

This composition was chosen because the homogeneous alloy is commercially available, and it also represents a reasonable Fe-Si endmember for data interpolation within 10 wt% light element in the core.

[17] 153: A 6000 K ICB was an assumption even in the referenced study. Perhaps briefly justify this temperature choice.

In the revision, we selected a preferred T_{ICB} value of 5,440 K based on the review by Hirose *et al.* (2013).

)

[18] line 154: The “M” in PREM already stands for “model”. Also, there have been recent revaluations of inner core properties (e.g., 10.1126/science.aav2296 and references therein), is PREM still the best point of comparison for this study?

The V_s proposed by Tkalčić, and Phạm (*Science*, 362, 329-332 (2018)) is about 1.1% smaller than the PREM value, and such errors are shown as the size of the shadow area in **Figure 6** in the revision. PREM still is the best point of comparison for this study.

[19] line 158: Perhaps similarity would be “not so surprising” but Fe and Fe-8.6Si appear identical. There are several studies which show at least a small effect of Si content on V_p (e.g., 10.1029/2003GL018405).

According to our new measurements V_p of Fe and Fe-8.6Si, we find the V_p of Fe-Si alloy almost linearly increases with the Si content at a constant density. We added **Supplementary Figure S10** to illustrate the point. We also added **Supplementary**

Figure S11 to show that the density of Fe-Si almost linearly decreases with the Si content at same pressure. As a combined result, the V_p of Fe-Si system is insensitive to the Si content under inner core conditions.

[20] line166: How much higher than that of Fe?

The σ of Fe-4.5Si is ~9% higher than that of Fe and ~ 11 % smaller than the PREM value.

[21] line170: How is “near melting” different than “pre-melting”? Is there any melting outside of the shaded “melting region”?

Thank you for the comment. The statement of “near melting” in line 170 is inaccurate. We revised sentence as “*Although we observed a considerable decrease in V_S and a considerable increase in σ during melting (Figs. 1 and 4), these drastic changes were caused by the effect in the solid-liquid mixing-phase region and not associated with pre-melting behavior*”

[22] 185: Mention that this is ~8 wt% C.

We have added the information.

[23]198: This sentence just repeats Line 190.

We have deleted the sentence.

[24] 201: The core C content is controversial, maybe this should be reframed as the required 4% inner core C requiring a certain bulk core or bulk Earth abundance.

Reworded the sentence (line 304-307).

[25] line 202: You should cite a study of solid-liquid alloy C partitioning and comment on whether it is reasonable for the inner core to have all the C.

We added the references in the revision (lines 267-271).

[26] line 208: Be more specific about the types of data that could help explore this problem.

We suggested specific experiments required to resolve the issues (lines 317-319).

[27]Figure 1: Please clarify how the boundaries of the “melt region” were determined and the associated pressures. I recommend using a different color for the Nguyen data and adjusting the “melting region” bar so that it does not cover the axes and tick marks.

The boundary of the melting region is determined according to the discontinuity of the sound velocity.

[28] Figure 2: Same as Figure 1.
Figure captions have been revised.

[29] Figure 3: Plotting the Si-only best match (Fe-3.6Si) would make it clearer why another light element is required. I recommend changing the way the PREM line is plotted (to a continuous line instead of unevenly spaced symbols) and labeling PREM on the line instead of in the corner. The properties of Fe-1Si-4C were interpolated from two other alloys (Fe₇C₃ and Fe-8.6Si) so how is it possible that Fe-1Si-4C has the smallest uncertainty of the three?

We incorporated some of the suggestions in **Figure 6** (Figure 3 in the original version) in the revision. Within the uncertainties, the composition of Fe-1Si-5C provides the best fits to the profiles of ρ , V_P , V_S and σ of the inner core. The errors of the density and sound velocity of Fe-1Si-5C is related to the weight percent of the Fe, Fe-8.6Si and Fe₇C₃ (see the **Method**), and the uncertainty for Fe-1Si-5C is not the smallest, comparing those of the endmembers.

[30] Figure S9: This figure appears to show a notable effect of Ni on Fe alloys which is not examined in the main text. Would the lines in Figure 3 change if they were recalculated with a realistic Ni component?

We considered this is an important figure and it has been moved to the main text (**Figure 5**). Figure 5 shows that the slope of the sound velocity of Fe-Ni is larger than that of pure iron. However, the data range is limited, and data are more scattered. The extrapolation is sensitive to the slope and so it is difficult to judge if the effect of Ni on the sound velocity is reliable when it is extrapolated to core pressures. We are hesitating to place too much weight to this dataset and making inference, but the sound velocity of Fe-Ni should be investigated under shock compression in the future.

REVIEWERS' COMMENTS

Reviewer #2 (Remarks to the Author):

The revised version of this study is a substantial improvement, and the authors have done a particularly good job at clarifying the motivation behind their alloy mixing approach. There are just a few points that should be clarified before this study is ready for publication. Also, there are some lingering grammatical mistakes; I recommend letting a native English speaker proofread the final manuscript.

L40 and elsewhere: Should be "Birch's law" not "the Birch's law"

L283: The authors should state that the best fit composition was chosen without considering the error bars of the endmembers, as explained in the reviewer response document.

L249: I am still unsure of what values were used to calculate the Fe₇C₃ properties. Table S3 has not changed in the revision and does not seem like it contains some parameters (like shear modulus, bulk modulus) that would be needed to calculate V_p , V_s , σ for a composition without measurements. All of the starting parameters and the actual equation used to calculate the derived (Table S4) values should be explicitly identified in the manuscript or supplement.

L432: What is this "assumed value"?

L435: Equations 10 and 11 should be explained. The reader shouldn't have to infer what the subscripts, delts, and deltas mean.

Figure 5: There should be a sentence in the main text mentioning the Ni effect and justifying not including it in the analysis, as explained in the reviewer response document.

Figure 6: The red circle and its error bars are hard to see and should be larger.

Responses to the reviewer's comments:

Reviewer #2 (Remarks to the Author):

The revised version of this study is a substantial improvement, and the authors have done a particularly good job at clarifying the motivation behind their alloy mixing approach. There are just a few points that should be clarified before this study is ready for publication. Also, there are some lingering grammatical mistakes; I recommend letting a native English speaker proofread the final manuscript.

Thanks for the positive review of our revision.

L40 and elsewhere: Should be "Birch's law" not "the Birch's law"

Corrected.

L283: The authors should state that the best fit composition was chosen without considering the error bars of the endmembers, as explained in the reviewer response document.

Added a clarification.

L249: I am still unsure of what values were used to calculate the Fe₇C₃ properties. Table S3 has not changed in the revision and does not seem like it contains some parameters (like shear modulus, bulk modulus) that would be needed to calculate V_p, V_s, σ for a composition without measurements. All of the starting parameters and the actual equation used to calculate the derived (Table S4) values should be explicitly identified in the manuscript or supplement.

The parameters needed for Fe₇C₃ density calculations are listed in Table S3. As explained in the text, we adopted the V_p and V_s data at room temperature (line 337) from Chen et al (2018). The V_p values for Fe₇C₃ were calculated according to Birch's law, whereas the V_s values were calculated by $V_s = V_s(300K, \rho) + (dV_s/dT)_\rho (T - 300)$, assuming the $(dV_s/dT)_\rho$ value of Fe₇C₃ is the same as that of pure iron. See modification in lines 338-340.

L432: What is this "assumed value"?

We assumed the (dV_s/dT) value for Fe₇C₃ is the same as that of pure iron (Fig. 3) (Line 340).

L435: Equations 10 and 11 should be explained. The reader shouldn't have to infer what the subscripts, delts, and deltas mean.

Modified as suggested.

Figure 5: There should be a sentence in the main text mentioning the Ni effect and justifying not including it in the analysis, as explained in the reviewer response document.

Added (lines 264-268).

Figure 6: The red circle and its error bars are hard to see and should be larger.

Done.